# PAM-flexible Cas9-mediated base editing of a hemophilia B mutation in induced pluripotent stem cells

Takafumi Hiramoto [1], Yuji Kashiwakura[1], Morisada Hayakawa [1,2], Nemekhbayar Baatartsogt [1], Nobuhiko Kamoshita[1,2], Tomoyuki Abe [3], Hiroshi Inaba [4], Hiroshi Nishimasu[5], Hideki Uosaki [3], Yutaka Hanazono[2,3], Osamu Nureki [6] & Tsukasa Ohmori [1,2✉]

## Abstract

**Background** Base editing via CRISPR-Cas9 has garnered attention as a method for correcting disease-specific mutations without causing double-strand breaks, thereby avoiding large deletions and translocations in the host chromosome. However, its reliance on the protospacer adjacent motif (PAM) can limit its use. We aimed to restore a disease mutation in a patient with severe hemophilia B using base editing with SpCas9-NG, a modified Cas9 with the board PAM flexibility.

**Methods** We generated induced pluripotent stem cells (iPSCs) from a patient with hemophilia B (c.947T>C; I316T) and established HEK293 cells and knock-in mice expressing the patient's *F9* cDNA. We transduced the cytidine base editor (C>T), including the nickase version of Cas9 (wild-type SpCas9 or SpCas9-NG), into the HEK293 cells and knock-in mice through plasmid transfection and an adeno-associated virus vector, respectively.

**Results** Here we demonstrate the broad PAM flexibility of SpCas9-NG near the mutation site. The base-editing approach using SpCas9-NG but not wild-type SpCas9 successfully converts C to T at the mutation in the iPSCs. Gene-corrected iPSCs differentiate into hepatocyte-like cells in vitro and express substantial levels of *F9* mRNA after subrenal capsule transplantation into immunodeficient mice. Additionally, SpCas9-NG–mediated base editing corrects the mutation in both HEK293 cells and knock-in mice, thereby restoring the production of the coagulation factor.

**Conclusion** A base-editing approach utilizing the broad PAM flexibility of SpCas9-NG can provide a solution for the treatment of genetic diseases, including hemophilia B.

## Plain language summary

In patients with hemophilia B, the blood does not clot properly, leading to excessive bruising and bleeding. Hemophilia B is caused by an error in a gene called coagulation factor IX (*F9*). To treat patients with hemophilia B, we might be able to use a technology called CRISPR-Cas9 to edit and correct this genetic error, restoring factor IX function and improving clotting. Here, we test a specific CRISPR-Cas9 approach in cells and animals. We show that we are able to correct the genetic error in *F9* in cells isolated from a patient with severe hemophilia B. We also show that we can fix the error in mice and that this increases levels of factor IX in the blood of the mice. With further testing, this gene-editing approach may be a viable therapy for patients with hemophilia B or similar genetic disorders.

[1] Department of Biochemistry, Jichi Medical University School of Medicine, 3311-1 Yakushiji, Shimotsuke, Tochigi 329-0498, Japan. [2] Center for Gene Therapy Research, Jichi Medical University, 3311-1 Yakushiji, Shimotsuke, Tochigi 329-0498, Japan. [3] Division of Regenerative Medicine, Center for Molecular Medicine, Jichi Medical University, 3311-1 Yakushiji, Shimotsuke, Tochigi 329-0498, Japan. [4] Department of Laboratory Medicine, Tokyo Medical University, 6-7-1 Nishishinjuku, Shinjuku-ku, Tokyo 160-0023, Japan. [5] Structural Biology Division, Research Center for Advanced Science and Technology, The University of Tokyo, 4-6-1 Komaba, Meguro-ku, Tokyo 153-8904, Japan. [6] Department of Biological Sciences, Graduate School of Science, The University of Tokyo, 7-3-1 Hongo, Bunkyo-ku, Tokyo 113-0033, Japan. ✉email: tohmori@jichi.ac.jp

Hemophilia is a congenital hemorrhagic disease caused by a genetic mutation in coagulation factor VIII (FVIII) or factor IX (FIX)[1]. Gene therapy holds an exciting prospect for patients and their families due to its potential to treat hemophilia via a single treatment. Recent clinical trials have established the efficacy of gene therapy for hemophilia using adeno-associated virus (AAV) vectors, which are considered to be a safe approach to gene therapy because the viral genome is maintained episomally.[2] Although AAV vectors enable long-term transgene expression in quiescent cells, cell division dilutes the AAV genome, resulting in the loss of therapeutic effect[3]. Hence, current hemophilia gene therapy with AAV vectors is only adapted to adult patients because their hepatocytes are mostly quiescent. Permanent repair of the gene responsible for hemophilia is a valuable goal for medical science.

We developed a gene-editing approach for hemophilia B using the CRISPR-Cas9 system to overcome the problems faced by current gene therapy techniques[4]. We inserted cDNA for exons 2–8 into intron 1 of *F9*. The therapeutic effects of genome editing persist permanently, even if the vector only exists transiently within liver cells. A phase I/II trial of the genome-editing treatment using AAV vectors harboring a zinc finger nuclease to insert cDNA of FIX into the *ALB* locus had been conducted[5]. These approaches to in vivo genome editing insert the cDNA at the site of a double-strand break (DSB) in a target locus. The disease-specific mutation cannot be effectively treated due to the lower frequencies of homology-directed repair (HDR)[6,7]. We previously attempted to specifically treat a disease-specific mutation by HDR; however, the frequency of HDR was <1%[4]. Creating DSBs by a DNA-cutting enzyme could generate large deletions and shuffle genes, which may result in adverse effects[8–11]. Therefore, an alternative approach to correct disease-specific mutations without DSBs would be advantageous.

Base editing is a novel genome-editing technology that can convert a specific DNA base into another at a targeted genome. Base editors are chimeric proteins that comprise a DNA-targeting module and a catalytic domain capable of deaminating a cytidine or adenine base (C to T or A to G)[12–14]. Cas9 nickase (that does not cause a DSB) could be used as a targeting module to bind specific DNA and deliver base editors to the mutation site[12–14]. Although several studies have reported the treatment of disease-specific mutations using base editing[14–20], PAM sequence requirement by Cas9 (NGG for SpCas9 or NNGRRT for SaCas9) hinders its application in various mutations. Thus, a more flexible module is required to bind the target DNA to apply the base-editing approach to various mutations. We recently developed engineered Cas9 (SpCas9-NG), which recognizes NGN (a single guanine) as the PAM sequence[21]. SpCas9-NG-mediated base editors are predicted to be applicable to a variety of mutations in human genetic diseases.

Here, we investigate whether base editing with SpCas9-NG can efficiently repair a point mutation of a patient with severe hemophilia B. We generate induced pluripotent stem cells (iPSCs) from a patient with severe hemophilia B (c.947T>C; I316T). The base editing with SpCas9-NG, but not wild-type SpCas9, successfully converts C to T at the mutation. The hepatocyte-like differentiated cells derived from gene-corrected iPSCs can express *F9* mRNA after the transplantation into immunodeficient mice. The base-editing approach using SpCas9-NG corrects the mutation in both HEK293 cells and knock-in mice with the mutation, thus restoring the production of the functional coagulation factor.

## Methods

**Generation of human iPSCs and cell culture**. Peripheral blood samples from a healthy donor and a patient with severe hemophilia B were collected after obtaining written informed consent, and experiments were approved by the ethical committee of Jichi Medical University (approved number: 18-18). iPSCs were generated as previously described with minor modifications[22]. Peripheral blood-derived mononuclear cells were cultured in StemFitAK02N (Ajinomoto Healthy Supply Co., Inc. Tokyo, Japan) without Supplement C, containing 50 ng/mL of human interleukin (IL)–6 (BioLegend, San Diego, CA, USA), 100 ng/mL of human stem cell factor (BioLegend), 10 ng/mL of human thrombopoietin (BioLegend), and 100 ng/mL human Fms-related tyrosine kinase ligand (BioLegend) for 3 days. Mononuclear cells were then transduced with Sendai virus vectors expressing reprogramming genes (CytoTune-iPS 2.0, ID Pharma Co., Tokyo, Japan). Four days after transduction, cells were harvested and seeded on plates coated with Corning Matrigel hESC-Qualified Matrix (Corning Inc., Corning, NY, USA) and cultured in StemFitAK02N. The medium was changed every other day until colonies appeared. Colonies were mechanically dissociated and seeded onto Matrigel-coated dishes. Established iPSCs were maintained in StemFitAK02N on plates coated with Matrigel or iMatrix-511 (Matrixome Inc., Osaka, Japan).

HEK293 cells were maintained in DMEM (Merck SA., Darmstadt, Germany) containing 10% fetal bovine serum (FBS; ThermoFisher Scientific, Waltham, MA, USA), GlutaMAX™ Supplement (ThermoFisher Scientific), and penicillin–streptomycin solution (FUJIFILM Wako Pure Chemical, Osaka, Japan).

**Plasmid construction and transduction**. cDNAs for SpCas9, SpCas9-NG, and SpCas9 (D10A) or SpCas9 (D10A) conjugated with the cytidine base editor PmCDA1 from sea lamprey (TAID) were provided by Prof. Nishimasu and Prof. Nureki (The University of Tokyo)[21]. Cytidine base editor APOBEC1 (BE4)-GAM[23] plasmid (#100806) was obtained from Addgene (Watertown, MA, USA). Plasmids were constructed using routine molecular cloning methods and confirmed by DNA sequencing.

Transduction to iPSCs was performed using P3 Primary Cell 4D-Nucleofector X Kit S (Lonza, Basel, Switzerland) as per the manufacturer's instructions. Briefly, $4 \times 10^5$ cells were suspended in a Nucleofector solution mix with 7 µg of plasmids. Cell suspension mixtures were then transferred into cuvettes, and the nucleofector process was initiated using a 4D-Nucleofector core unit (Lonza, Basel, Switzerland). After electroporation, cells were seeded on iMatrix-511-coated wells and cultured in StemFitAK02N with 10 µM Y-27632 (FUJIFILM Wako).

To establish HEK293 cells stably expressing a human *F9* cDNA, the linearized plasmid (pBApo-Neo-EF1α, Takara Bio, Shiga, Japan) expressing *F9* cDNA (R338L, R338L+I316T patient mutation) was transfected into the cells with Lipofectamine 3000 Reagent (ThermoFisher Scientific), according to the manufacturer's instructions. G418 (400 µg/mL) (Nacalai Tesque Inc., Kyoto, Japan) was added to the culture medium to obtain stably expressed clones.

**Surveyor endonuclease assay**. Genomic DNA was extracted using the DNeasy Blood & Tissue kit (QIAGEN, Venlo, Netherlands). PCR fragments were amplified using ExTaq DNA polymerase (Takara Bio), then denatured and re-annealed using a thermal cycler. Genomic mutations were detected using the Surveyor Mutation Detection Kit (Integrated DNA Technologies, Skokie, IL). DNA fragments were analyzed by agarose gel electrophoresis. The oligonucleotide primer sequences used to detect mutations are described in Supplementary Table 1.

**Next-generation sequencing**. Genomic DNA fragments were amplified using Prime Star MAX polymerase (Takara Bio) and

purified using AMPure XP beads (Beckman Coulter, Brea, CA). A library was prepared to add adapters and a barcode sequence to the amplicon target. PCR amplicons were subjected to 250 or 300 pair-end read sequencing using Illumina MiSeq (100,000 reads) at Research Institute for Microbial Diseases at Osaka University (Osaka, Japan) or Hokkaido System Science (Hokkaido, Japan). The primer sequences are described in Supplementary Table 1.

**Immunohistochemical staining.** The immunohistochemical staining of iPSCs was performed to confirm the undifferentiated state[24]. Human iPSCs cultured in 12-well plates were fixed with 4% paraformaldehyde in phosphate-buffered saline (PBS) at 4 °C for 30 min and permeabilized with PBS containing 0.1% Triton X-100. Sections were blocked with 5% skim milk in PBS containing 0.1% Tween-20 (PBS-T) for 30 min at 4 °C. Cells were stained with anti-NANOG mouse monoclonal antibody (10 μg/mL, clone 23D2-3C6; BioLegend) and anti-OCT4 mouse monoclonal antibody (2.5 μg/mL, clone 3A2A20; BioLegend) at 4 °C overnight. Antibody binding was detected using anti-mouse IgG conjugated with Alexa Fluor 594 (2 μg/mL, ThermoFisher Scientific). Sections were mounted in VECTASHIELD Mounting Medium with DAPI (Vector Laboratories, Burlingame, CA, USA). Immunofluorescence staining was observed by using an all-in-one microscope (BIOREVO BZ-9000, KEYENCE, Tokyo, Japan).

For the immunohistochemistry of HEK293 cells, the cells were fixed with 4% paraformaldehyde for 20 min at room temperature and were washed with PBS containing 0.1% Tween-20. The cells were permeabilized with PBS containing 0.3% Triton X-100 and 2% goat serum for 1 h, and they were subsequently incubated with goat anti-human FIX polyclonal antibody conjugated with biotin (10 μg/mL, GAFIX-AP, Affinity Biologicals, Ancaster, ON, Canada) and rabbit anti-Giantin polyclonal antibody (1 μg/mL, Poly19087; BioLegend) overnight at 4 °C. Next, the binding antibodies were visualized using streptavidin conjugated with Alexa Fluor 594 and anti-rabbit antibody conjugated with Alexa Fluor 488 (10 μg/mL, ThermoFisher Scientific) for 1 h at room temperature. Slides were mounted with VECTASHIELD mounting medium with DAPI. Immunofluorescence staining was observed and photographed using a confocal microscope (Leica TCS SP8; Leica Microsystems, Wetzlar, Germany).

**Flow cytometry analysis.** Cells were suspended in PBS containing 0.5% bovine serum albumin and 2 mM EDTA and incubated with R-Phycoerythirin-conjugated anti-human SSEA-4 monoclonal antibody (2 μg/$10^6$ cells, clone MC-813-70; BioLegend), and Alexa Fluor 488-conjugated anti-human Tra-1-60-R monoclonal antibody (2 μg/$10^6$ cells, clone TRA-1-60-R; BioLegend). Cells were resuspended using the BD Cell Viability kit (2 μg/$10^6$ cells, BD Bioscience, Franklin Lakes, NJ). The surface expression of target proteins in living cells was analyzed on an LSRFortessa X-20 (BD Bioscience). FCS files were obtained using Diva software and re-analyzed using FlowJo software (BD Bioscience).

**Mice.** NOG (NOD.Cg-Prkdcscid Il2rgtm1Sug/Jic) and TK-NOG [NOD.Cg-Prkdcscid Il2rgtm1SugTg (Alb-UL23)7-2/ShiJic] mice were purchased from In-Vivo Science Inc. (Tokyo, Japan). Knock-in (KI) mice expressing human *F9* cDNA without or with the hemophilia B mutation (I316T) [C57BL/6-*F9*$^{tm1.1(hF9\_R338L)}$ and C57BL/6-*F9*$^{tm1.2(hF9\_R338L\_I316T)}$] were developed by UNITECH (Chiba, Japan). Briefly, a linearized targeting vector containing a human *F9* minigene (exon 1, truncated intron 1, and exon 2–8) (R338L mutation or R338L+I316T mutations), SV40 polyA sequence, and PGK-Neo cassette with 5′ arm (4.3 kb) and 3′ arm (2.3 kb) was transduced into mouse embryonic stem cells

(ESGRO Complete™ Adapted C57/BL6 mouse embryonic stem cell line, Merck, Darmstadt, Germany) by electroporation. The embryonic stem cells were then injected into mouse blastocysts after confirming gene targeting by Southern blotting. Chimeric mice were bred with C57BL/6 mice to produce *F1* mice and then bred with CAG-FLPe transgenic mice to delete the PGK-Neo cassette. All animals were housed in groups of a maximum of five mice in individually ventilated cages under specific pathogen-free animal facilities. Dark/light cycles were 12 h/12 h with a temperature of 23 °C and humidity of 50%. All experiments were performed in compliance with the Institutional Animal Care and Concern Committee at Jichi Medical University, and animal care was carried out in accordance with the committee's guidelines.

**Teratoma formation and histological analysis.** iPSCs ($1 \times 10^6$ cells) were injected into the testis of immunodeficient NOG male mice (6-week-old). At 12–17 weeks post-injection, mice were anesthetized with isoflurane and perfused with PBS. Tumor tissues were then fixed with 10% formalin. Paraffin-embedded tissue sections were dewaxed using xylene and rehydrated using ethanol and water. Sections were processed for hematoxylin and eosin staining and then visualized using an all-in-one microscope (BIOREVO BZ-9000, KEYENCE).

**Differentiation of iPSCs into hepatic cell lineages.** Hepatic cell lineage differentiation was performed in vitro as previously described with minor modification[25,26]. iPSCs were dissociated into single cells using TrypLE-Select (ThermoFisher Scientific) and plated onto Matrigel Growth Factor Reduced Basement Membrane Matrix (Corning). Cells were maintained in StemFitAK02N for a few days until 80% confluent. Endoderm differentiation was initiated using RPMI1640 medium supplemented with 50 ng/mL of human WNT3A (R&D systems, Minneapolis, MN), 100 ng/mL of human Activin A (BioLegend), and B-27 Supplement (ThermoFisher Scientific) for 4 days. Induction of hepatocyte differentiation was performed by culturing the endoderm cells in RPMI1640 supplemented with 20 ng/mL of fibroblast growth factor 4 (BioLegend), 20 ng/mL of bone morphogenetic protein 4 (BioLegend), and B-27 Supplement for 5 days, followed by culture in RPMI1640 supplemented with 20 ng/mL of human hepatocyte growth factor and B-27 Supplement in hypoxic conditions using Hypoxia Chamber (STEMCELL Technologies Inc., Vancouver, BC, Canada). Cells were further cultured for 11 days in Hepatocyte Culture Media (Hepatocyte Culture Media Bullet kit, Lonza, Basel, Switzerland) supplemented with 20 ng/mL of human Oncostatin M (FUJIFILM Wako Pure Chemical).

**Subrenal capsule implantation of iPSC-derived hepatic cells into TK-NOG mice.** Engraftment of differentiated iPSCs was performed using TK-NOG mice[26]. Eight-week-old TK-NOG male mice were injected intraperitoneally with 6 mg/kg ganciclovir at 8 and 6 days prior to transplantation. Differentiated iPSC-derived cells ($2 \times 10^6$) were suspended in ice-cold Hepatocyte Culture Media (Lonza) and Matrigel Growth Factor Reduced Basement Membrane Matrix and then transplanted in the subrenal capsule of TK-NOG mice anesthetized with isoflurane. Mouse kidneys were harvested 12 weeks after transplantation, and mRNA expression of *F9* was analyzed. For the detection of the FIX antigen in engrafted tissue via immunohistochemistry, tissue samples were fixed with 4% paraformaldehyde, incubated with PBS containing sucrose, and then frozen in Tissue-Tek O.C.T. Compound (Sakura Fintek Japan, Tokyo, Japan). The sections were incubated with 2% goat serum and 0.3% Triton X-100 and further treated with Ready Probes Streptavidin/Biotin

Blocking Solution (Invitrogen, USA). The sections were then incubated with goat-human FIX polyclonal antibody conjugated with biotin (10 μg/mL, GAFIX-AP, Affinity Biologicals) overnight at 4 °C. The antibody binding was detected with streptavidin conjugated with Alexa Fluor 594 (10 μg/mL, ThermoFisher Scientific). Slides were mounted with VECTASHIELD mounting medium with DAPI. Immunofluorescence staining was observed with a Leica SP8 confocal microscope (Leica Microsystems).

**Quantitative reverse transcription polymerase chain reaction**. Total RNA was isolated using the RNeasy Mini Kit (QIAGEN). Human liver RNA was purchased from Takara Bio. cDNA was synthesized using the PrimeScript RT-PCR Kit (Takara Bio). Quantitative polymerase chain reaction (PCR) was performed using THUNDERBIRD Probe qPCR Mix (Toyobo, Osaka, Japan) or THUNDERBIRD SYBR qPCR Mix (TOYOBO) on QuantStudio 12K Flex (ThermoFisher Scientific). Reactions were analyzed in duplicate and expression levels were normalized to *GAPDH* mRNA levels. Predesigned primer and probe sets were used for the experiments (TaqMan Gene Expression Assays, ThermoFisher Scientific, *AFP*: Hs00173490, *ALB*: Hs00910225, *CYP3A4*: Hs00604506, *HNF4A*: Hs00230853, *F9*: Hs01592597, *GAPDH*: Hs02758991). The predesigned primers for SYBR qPCR were purchased from Takara Bio (*F9*: HA166488 *GAPDH*: HA067812).

**Measurement of FIX activity and FIX antigen**. The medium of confluent cells was changed into a fresh medium containing 5 μg/mL of menatetrenone (Eisai, Tokyo, Japan), and the culture was continued for the following 48 h. Blood sample was drawn from the jugular vein of anesthetized mice with a 29G micro-syringe (TERUMO Corp., Tokyo, Japan) containing 1/10 (volume/volume) sodium citrate. FIX activity (FIX:C) was measured with a one-stage clotting-time assay using an automated coagulation analyzer (Sysmex CA-1600 analyzer; Sysmex, Kobe, Japan). The FIX antigen (FIX:Ag) was measured as follows: microtiter plates were incubated with PBS containing anti-human FIX antibody (CL20039AP; CEDARLANE, Ontario, Canada) (1 μg/mL) overnight at 4 °C. After blocking with 5% casein in PBS for 1 h at room temperature, samples were incubated in PBS containing 1% casein and 0.1% Triton X-100 for 1 h at 37 °C. After washing with PBS containing 0.1% Triton X-100, the bound FIX antigen was detected with the anti-human FIX antibody conjugated with horseradish peroxidase (Affinity Biologicals) using ABTS microwell peroxidase substrate (Seracare, Milford, MA, USA).

**AAV vector construction and vector administration**. The SpCas9 cannot be packaged in one AAV vector; thus, we employed an intein-mediated split-Cas9 system[27]. The SpCas9-base editor was expressed by two AAV vectors. To express the target gene specifically in hepatocytes, we created a plasmid that contains a chimeric promoter (HCRhAAT; an enhancer element of the hepatic control region of the Apo E/C1 gene and the human anti-trypsin promoter), cDNAs, and the SV40 polyadenylation signal. The AAV genes were packaged by triple plasmid transfection of AAVpro293T cells (Takara Bio) to generate the AAV vector[4]. We used the AAV8 serotype, as it efficiently expresses the target gene in the mouse liver in vivo. The AAV vector was quantified by qPCR measuring the SV40 polyadenylation signal. The primer pairs and probe were as follows:

5′-AGCAATAGCATCACAAATTTCACAA-3′ (sense)
5′-CCAGACATGATAAGATACATTGATGAGTT-3′ (anti-sense)
5′-AGCATTTTTTTCACTGCATTCTAGTTGTGGTTTGTC-3′ (FAM probe)

For the in vivo base-editing experiments, we administered two AAV vectors expressing the N- and C terminal of SpCas9-base

editor intraperitoneally (10 μL) in neonatal mice ($1 \times 10^{11}$ viral genomes).

**GUIDE-Seq**. GUIDE-Seq was carried out as previously described, with minor modifications[28]. HEK293 cells were transfected with a plasmid expressing SpCas9 or SpCas9-NG and gRNA3 for 7 days together with dsODN. Genomic DNA was isolated, and then the libraries were prepared from 400 ng of genomic DNA using the NEBNext Ultra II FS DNA Library Prep Kit for Illumina (New England Biolabs, Ipswich, MA, USA) according to the manufacturer's instructions. Next, they were subjected to 150 pair-end read sequencing using the Illumina NovaSeq platform (50,000,000 reads) at the Genome Information Research Center at Osaka University. Data analysis was performed using the GUIDE-Seq analysis pipeline following standard procedures[29]. The primer sequences are described in Supplementary Table 1, and the in silico predicted off-target sites are listed in Supplementary Data 1.

**Statistics and reproducibility**. Statistical analyses were performed using Prism 9 software (Graph Pad Software, San Diego, CA, USA). All data are presented as mean ± standard error of the mean (SEM). Each data was obtained from biologically independent experiments (Supplementary Data 2). The sample number of the experiment was indicated in each figure legend. When indicated, statistical significance was determined using a two-tailed Student's *t*-test or one-way ANOVA with post hoc Tukey's multiple comparison test. All P-values <0.05 were considered statistically significant.

**Reporting summary**. Further information on research design is available in the Nature Portfolio Reporting Summary linked to this article.

## Results

**SpCas9-NG showed broad PAM flexibility in iPSCs**. We generated iPSCs from a patient with severe hemophilia B. Peripheral blood mononuclear cells from the patient were transduced using a Sendai virus vector harboring the four reprogramming factors. We confirmed that the patient-derived iPSCs had a missense mutation at exon 8 in the *F9* gene (c.947T>C) (Fig. 1a). The point mutation may be treated by cytidine base editors comprising Cas9 and deaminases such as PmCDA1 (TAID) and APOBEC1 (BE4), which mediate C to T conversion[13,14].

We found three possible PAM sites at the 3′ end of the mutation and designed three guide RNAs (gRNAs) to repair the mutation by a base-editing approach (Fig. 1b). We then inserted each gRNA downstream of the U6 promoter in the plasmid harboring wild-type SpCas9 or SpCas9-NG driven by CAG promoter (Fig. 1c). Using Surveyor endonuclease assay, we assessed a DSB near the mutation site induced by the expression of each gRNA and Cas9 after transduction of patient-derived iPSCs with each plasmid. SpCas9 induced DSBs with gRNA2 (TGG PAM) but not with gRNA1 (TGG PAM) or gRNA3 (CGA PAM). (Fig. 1d and Supplementary Fig. 1). In contrast, SpCas9-NG induced DSBs with all three gRNAs, albeit a low efficiency with gRNA1 (Fig. 1d and Supplementary Fig. 1). We quantified the frequency of DSBs induced by Cas9 with gRNA2 and gRNA3 by next-generation sequencing. SpCas9 induced DSBs at the TGG PAM site with gRNA2 more efficiently than SpCas9-NG (SpCas9, 8.72 ± 1.29%; SpCas9-NG, 3.16 ± 0.71%) (Fig. 1e). Notably, SpCas9-NG, but not SpCas9, induced DSBs at the CGA PAM site with gRNA3 efficiently (SpCas9, 0.21 ± 0.04%; SpCas9-NG, 6.55 ± 2.13%) (Fig. 1e). These results confirmed the broad PAM flexibility of SpCas9-NG in iPSCs, as observed in HEK293 cells. To evaluate off-target events in the whole genome, we performed GUIDE-Seq in HEK293 cells transducing the plasmid expressing

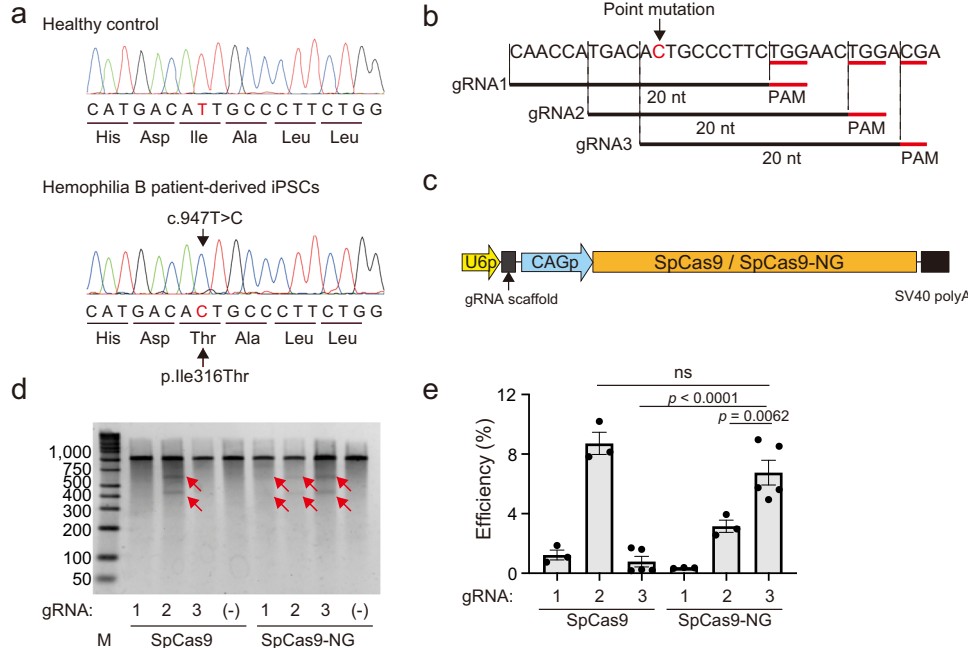

**Fig. 1 PAM flexibility of SpCas9-NG in a hemophilia B patient-derived iPSC. a** DNA sequencing of *F9* (part of exon 8) in control or severe hemophilia B patient-derived iPSCs. The patient-derived iPSCs contain the missense mutation (c.947T>C), leading to an amino-acid substitution (p.Ile316Thr). The replaced nucleotide is highlighted in red. **b** Schematic presentation of 20-nt guide RNA near the missense mutation. PAM sequences for each guide RNA are indicated in red. **c** Schematic presentation of the plasmid constructs. U6p, U6 promoter; CAGp, CAG promoter. **d, e** Patient-derived iPSCs were transduced using a plasmid expressing SpCas9 or SpCas9-NG and each gRNA. **d** Detection of induced DSBs using Surveyor endonuclease assay, following 2% agarose gel electrophoresis. Red arrows indicate the fragments caused by the Surveyor endonuclease digestion. M, DNA ladder marker. **e** Examination of the frequency of DSBs using next-generation sequencing. Values represent mean ± SEM ($n = 3$). Statistical significance was determined using one-way ANOVA with post hoc Tukey's multiple comparison test. ns, not significant.

SpCas9 or SpCas9-NG and gRNA3. The peak score was significantly higher for SpCas9-NG at one guide RNA mismatch, which corresponds to the patient's point mutation (Supplementary Fig. 2a, b and Supplementary Data 1). We could not detect any difference between the other scores and influenced genes of SpCas9 and those of SpCas9-NG (Supplementary Fig. 2a, b and Supplementary Data 1).

**Genetic correction of the mutation in patient-derived iPSCs by base editing.** We sought to correct the mutation (c.947T>C) using cytidine base editors comprising the D10A nickase version of Cas9 (SpCas9 or SpCas9-NG) conjugated with TAID or BE4 (Fig. 2a). We transduced the patient-derived iPSCs with a plasmid coding one of the four base editors and the gRNA (gRNA2 or gRNA3), and then measured the C to T correction at the mutation site using next-generation sequencing. TAID-conjugated SpCas9-NG (TAID-SpCas9-NG) with gRNA3 corrected the mutation most effectively, whereas we could not detect efficient base editing by gRNA3 with BE4-conjugated SpCas9 (BE-SpCas9) (Fig. 2b). Although SpCas9 with gRNA2 showed the highest DSB frequency (Fig. 1e), SpCas9 conjugated TAID (TAID-SpCas9) was less effective than TAID-SpCas9-NG (Fig. 2b). Furthermore, the efficiencies of C to T conversion by the BE4-SpCas9 with gRNA2 were lower than with the TAID-SpCas9-NG with gRNA3.

We next cloned iPSC colonies after the transduction of the patient-derived iPSCs with plasmids harboring SpCas9-NG-TAID with gRNA3. We detected 84 colonies and selected one that had the gene correction at the mutation (Fig. 2c). The isolated gene-corrected iPSCs could be maintained in an undifferentiated status since the cells expressed pluripotency markers (NANOG, OCT4, SSEA-4, and Tra-1-60) (Fig. 2d). Furthermore, the gene-corrected

iPSCs developed teratomas, representing differentiation into three germ layers, in NOG mice (Fig. 2e).

**Differentiation of patient-derived gene-corrected iPSCs into hepatic cells.** The gene-corrected iPSCs were differentiated into hepatic lineage cells to examine whether the cells could express FIX (Fig. 3a). The hepatic differentiation of the cells at day 25 was supported by an increased expression of hepatocyte-specific mRNAs, including *alpha-fetoprotein (AFP)*, *albumin (ALB)*, *cytochrome p450 3a4 (CYP3A4)*, and *hepatocyte nuclear factor 4 alpha (HNF4A)* (Fig. 3b). In contrast, expression of *F9* mRNA was marginal compared with that of other genes (Fig. 3b). Also, we differentiated the parental iPSCs into hepatic lineage cells and found the same levels of hepatic specific gene expressions as gene-corrected cells (Supplementary Fig. 3). The expressions of *AFP*, *ALP*, and *CYP3A* in the cells that were differentiated in vitro were comparable to, or greater than, those in human liver RNA, while the expressions of *HNF4A* and *F9* were lower than those in human liver RNA (Supplementary Fig. 3), suggesting incomplete differentiation. To further differentiate the cells, in vitro differentiated cells were transplanted into the subrenal capsule of TK-NOG mice treated with ganciclovir (Fig. 3a). The subrenal capsule transplantation procedure is commonly used to graft human cancer cells, while several reports observed the long-term engraftment of the transplanted hepatocytes or the differentiated cells from iPSCs[30–32]. We employed the previous procedure for the differentiation of iPSCs and transplantation of the cells[26]. At 12 weeks after transplantation, white engrafted tissue was observed in the kidneys (Fig. 3c). We observed robust *F9* mRNA expression in the engrafted tissues compared with in vitro differentiated cells (Fig. 3d). We also confirmed the expression of FIX protein in

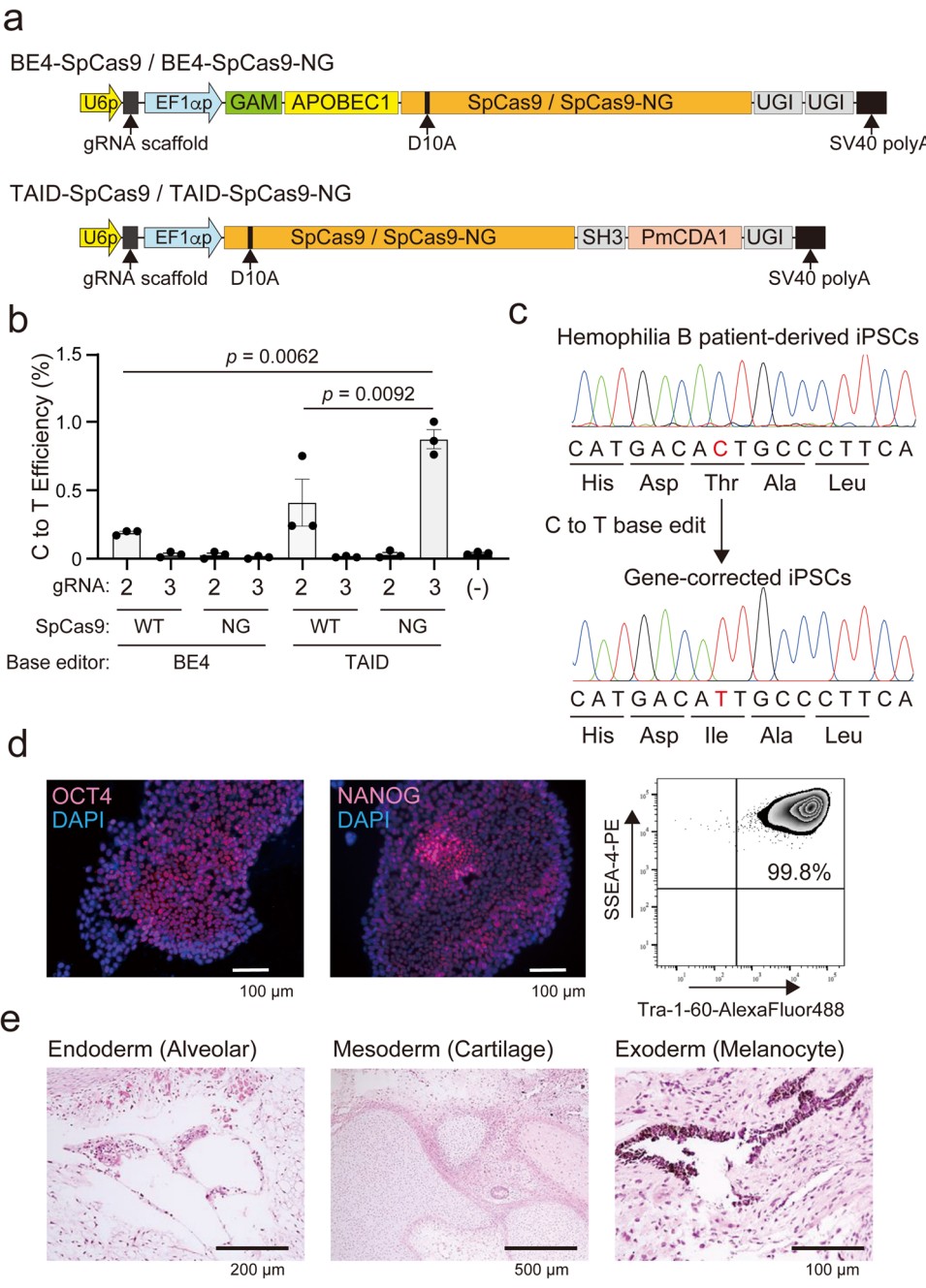

**Fig. 2 Genetic correction of hemophilia B patient-derived iPSCs by base-editing technology. a** Schematic presentation of the plasmid constructs for the cytidine base editor, SpCas9 conjugated BE4 (BE4-SpCas9), SpCas9-NG conjugated BE4 (BE4-SpCas9-NG), SpCas9 conjugated TAID (TAID-SpCas9), or SpCas9-NG conjugated TAID (TAID-SpCas9-NG). U6p, U6 promoter; EF1αp, EF1α promoter; GAM, bacteriophage Mu Gam protein; UGI, a uracil DNA glycosylase inhibitor. **b** The frequencies of C to T conversion at the mutation of the severe hemophilia B patient-derived iPSCs after transduction with BE4-SpCas9, BE4-SpCas9-NG, TAID-SpCas9, or TAID-SpCas9-NG plasmid harboring gRNA2 or gRNA3. The frequency of C to T conversion was examined using next-generation sequencing. Values represent mean ± SEM ($n = 3$). Statistical significance was determined using one-way ANOVA with post hoc Tukey's multiple comparison test. **c** DNA sequencing of *F9* (part of exon 8) in the severe hemophilia B patient-derived iPSCs or the gene-corrected iPSCs by the base editor. The replaced nucleotide is highlighted in red. **d** The expression of pluripotent markers in the patient-derived gene-corrected iPSCs. The expression of OCT4 and NANOG was examined by immunohistochemical staining. Sections were observed using an all-in-one microscope (BIOREVO BZ-9000). Red, OCT4 and NANOG; blue, DAPI. SSEA-4 and Tra-1-60 expression was determined by flow cytometry. **e** Sections from a gene-corrected iPSC-derived tumor in immune-deficient mice were stained with hematoxylin-eosin staining and then observed. Differentiation into three germ layers (endoderm, mesoderm, and exoderm) was confirmed.

the engrafted tissue via immunohistochemistry (Fig. 3e). However, we could not detect human FIX antigen in the mice plasma, and the expression levels of *F9* did not reach those observed in normal human liver (in vivo differentiated cells, $6.6 \pm 1.1 \times 10^4$; normal human liver, $86.6 \pm 9.9 \times 10^4$).

**Increase in FIX activity after gene correction by base-editing technology.** To confirm the restoration of FIX production by the SpCas9-NG-mediated base editing, we established the HEK293 cells stably expressing human *F9* cDNA and the patient's *F9* cDNA (I316T mutation). We inserted the R338L Padua mutation

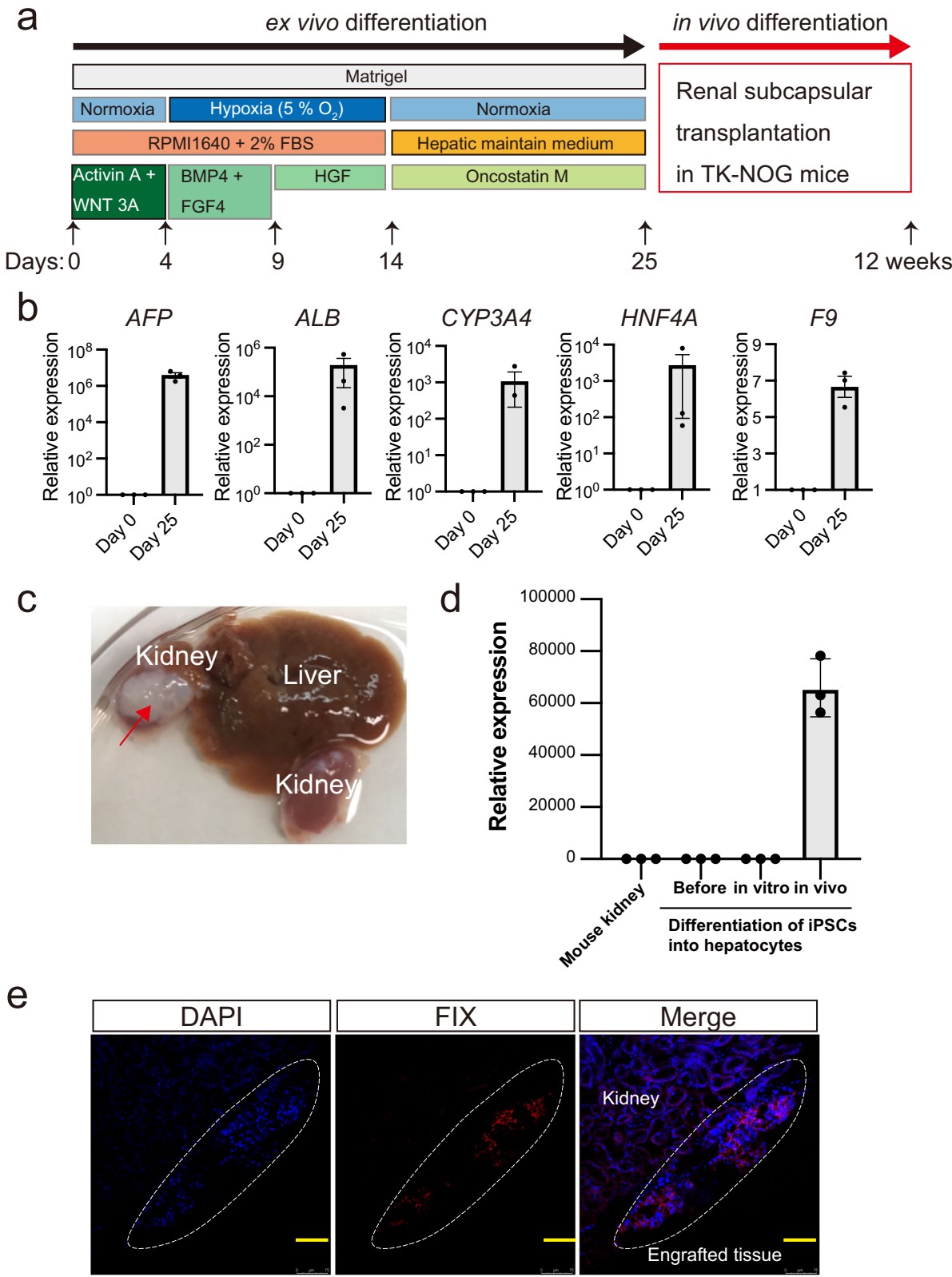

in *F9* cDNA to easily detect the increase in FIX activity by base editing. The activity of the *F9* R338L Padua mutation was eight times higher compared to that of wild-type *F9*[33]. HEK293 expressing *F9* with the I316T mutation expressed *F9* mRNA but did not produce the FIX antigen or activity in the culture media (Fig. 4a–c). Immunological staining showed the presence of FIX protein with the I316T mutation in the cells, suggesting that the

mutant FIX could not be released to the supernatant (Fig. 4d and Supplementary Fig. 4). Subsequently, we repaired the I316T mutation by transducing the cells that expressed it with TAID-SpCas9-NG and gRNA3. After the transduction, we cloned single-cell derived transduced cells, and we measured the FIX activity in the culture medium (Fig. 4e), which increased more than five times in eight clones out of 137 (the average of eight

**Fig. 3 Differentiation of gene-corrected hemophilia B patient-derived iPSCs into hepatic lineage cells. a** Schematic presentation of the procedure for hepatic lineage differentiation of iPSCs. iPSCs were differentiated in vitro for 25 days and then transplanted into the renal subcapsule of immune-deficient mice. **b** Relative mRNA expression of hepatocyte-specific proteins (*AFP, ALB, CYP3A4, HNF4A,* and *F9*) in the gene-corrected iPSCs before (day 0) and after hepatic lineage differentiation in vitro (day 25). Values represent mean ± SEM (*n* = 3). **c** Engraftment of the differentiated iPSCs in mouse kidney at 12 weeks after renal subcapsule transplantation in immune-deficient mice (red arrow). **d** Relative mRNA expression of *F9* in mouse kidney, gene-corrected undifferentiated iPSCs (before), hepatic lineage-differentiated gene-corrected iPSCs in vitro (in vitro), and engrafted tissues derived from hepatic lineage-differentiated gene-corrected iPSCs (in vivo). The same samples in (**b**) were analyzed for relative expressions of *F9* (Before, Day 0 in (**b**); in vitro, Day 25 in (**b**)). Values represent mean ± SEM (*n* = 3). **e** FIX expression in the gene-corrected iPSC-derived tissue was assessed by immunohistochemical staining. The staining was observed using a confocal microscope (Leica TCS SP8). Blue, DAPI; Red, FIX. Scale bars (75 μm) are shown in yellow.

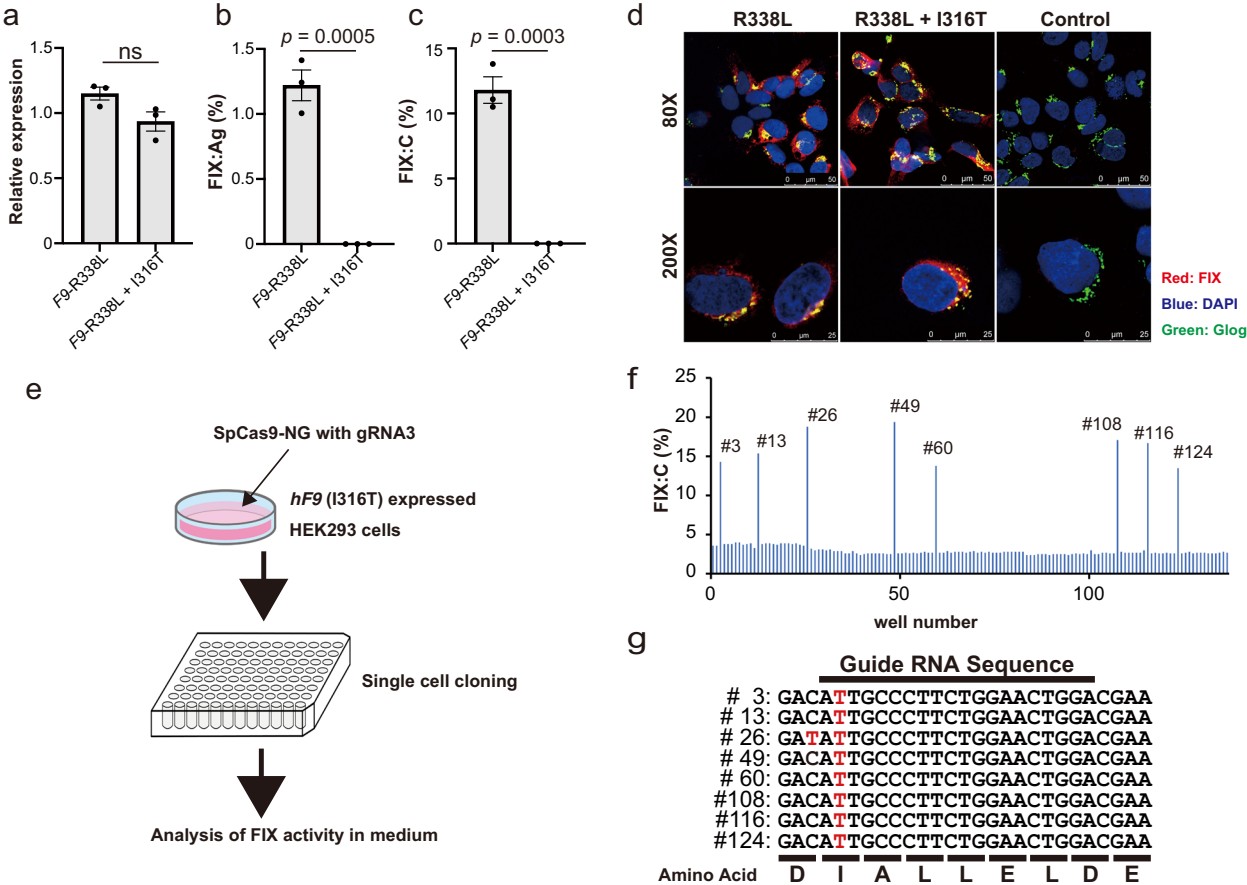

**Fig. 4 Restoration of FIX production by base-editing technology in HEK293 cells expressing the patient's mutation.** The HEK293 cells stably expressing human *F9* cDNA (R338L mutation) and the patient's *F9* cDNA (R338L+I316T mutations) were established by plasmid transfection. **a–c** Relative *F9* mRNA expression (**a**), FIX antigen (FIX:Ag) (**b**), and FIX activity (FIX:C) (**c**) in the supernatant were assessed. Values represent mean ± SEM (*n* = 3). ns, not significant. Statistical significance was determined using a two-tailed Student's *t*-test. **d** Intracellular FIX protein expression in the HEK293 cells stably expressing human *F9* cDNA (R338L mutation) and the patient's *F9* cDNA (R338L+I316T mutations) was assessed by immunohistochemical staining. Immunofluorescence staining was observed and photographed using a confocal microscope (Leica TCS SP8). Red, FIX; green, Golgi; blue, DAPI. The scales are indicated in the figures. **e** Schematic presentation of the isolation of cell clones after base editing. We transfected the plasmids (TAID-SpCas9-NG with gRNA3) into HEK293 cells expressing the patient's *F9* cDNA (R338L+I316T mutations). We cloned single-cell-derived colonies and evaluated the FIX:C in the medium. **f** Increase in FIX:C in the medium observed after transfection in eight out of 137 clones. **g** *F9* cDNA sequencing near the mutation in the clones showing increased expression of FIX:C in the medium. The red letters indicate the bases that were different from the DNA sequences obtained from the original cells.

clones was 16.1 ± 2.3%; that of the other clones was 2.9 ± 0.5%) (Fig. 4f). We confirmed that the mutation was repaired in the clones with elevated FIX activity (Fig. 4g). On the other hand, we could not obtain any clone with increased FIX activity in the supernatant in the cells transducing TAID-SpCas9 and gRNA3 (none of 30 clones). These results suggest that base editing with SpCas9-NG repairs the mutation of hemophilia B and rescues FIX production.

Finally, we investigated whether the base-editing technology could increase plasma FIX in vivo. We created knock-in (KI) mice that expressed human *F9* cDNA without or with the hemophilia B mutation (I316T) (Fig. 5a). We inserted the R338L mutation to *F9* cDNA to easily detect the increase in FIX activity, as in the experiments shown in Fig. 4. The plasma FIX antigens and activity in heterozygous KI mice expressing human *F9* cDNA without the mutation were 2.7 ± 1.8% and

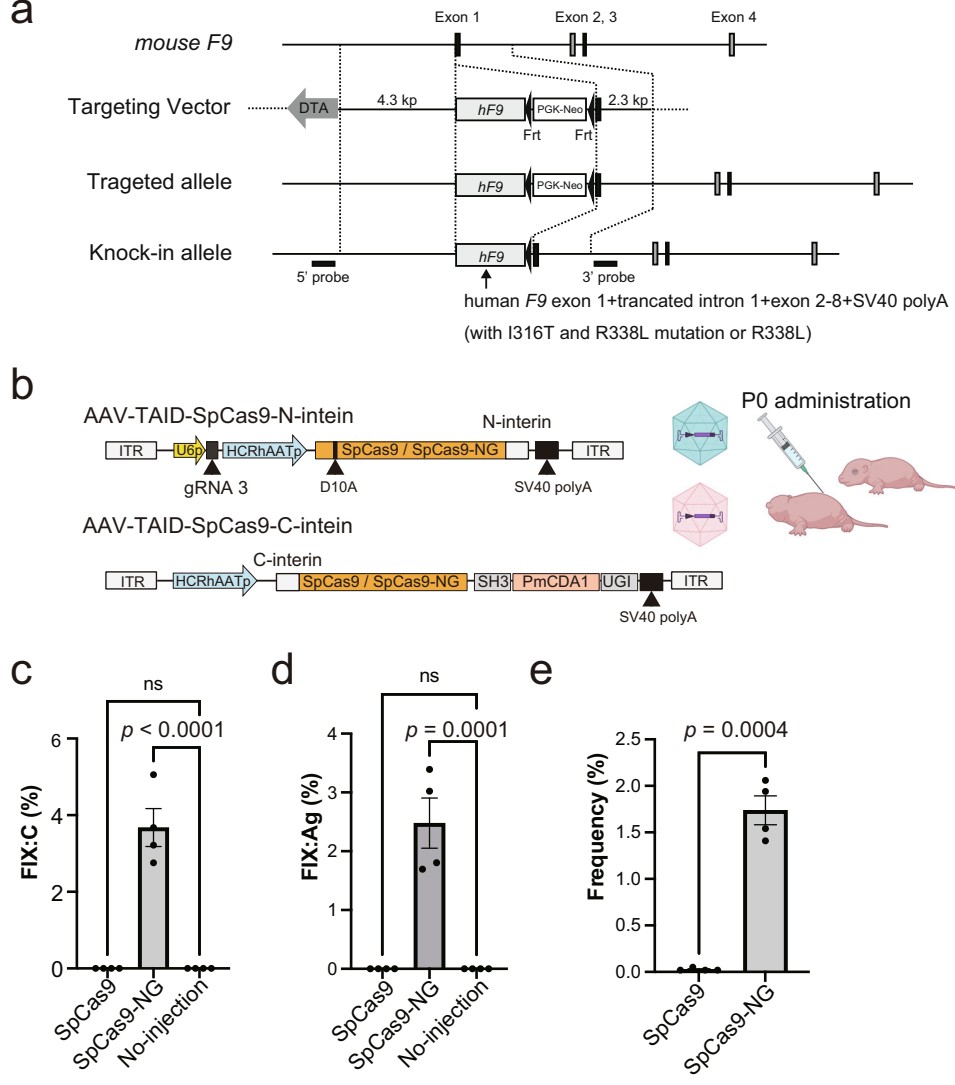

**Fig. 5 Restoration of FIX production in knock-in mice expressing the patient's *F9* cDNA by base-editing technology. a** Schematic presentation of the knock-in allele. **b** Schematic presentation of AAV vectors for intein-mediated split-Cas9 system expressing TAID-SpCas9 or TAID-SpCas9-NG. We injected two AAV vectors for the expression of TAID-SpCas9 or TAID-SpCas9-NG, as well as gRNA3, into the newborn pups expressing the patient's *F9* cDNA. U6p, U6 promoter; HCRhAATp, a chimeric promoter containing an enhancer element of the hepatic control region of the *Apo E/C1* gene and the human anti-trypsin promoter; UGI, a uracil DNA glycosylase inhibitor. **c, d** The increase in plasma FIX antigen (FIX:Ag) (**c**) and FIX activity (FIX:C) (**d**) at 6 weeks after the vector injection. The data were shown as the percentage of heterozygous knock-in mice expressing human *F9* cDNA without the I316T mutation. Values represent mean ± SEM (*n* = 4). ns, not significant. Statistical significance was determined using one-way ANOVA with post hoc Tukey's multiple comparison test. **e** The frequencies of C to T conversion at the mutation of the livers in the treated mice were assessed with next-generation sequencing. Values represent mean ± SEM (*n* = 4). Statistical significance was determined using a two-tailed Student's *t*-test. The illustration of (**b**) was created with BioRender.com.

21.8 ± 8.4%, respectively. However, we did not detect human FIX antigens and activities in the KI mice with the patient's mutation (Fig. 5c, d). To express the base editors in the liver of the KI mouse, we applied a dual AAV vector system using protein trans-splicing by intein[27] because the length of the SpCas9-base editor exceeds the packaging capacity of the AAV vector (Fig. 5b). We administrated the two AAV8 vectors intraperitoneally into newborn pups of KI mice with the patient's mutation. Plasma levels of FIX were significantly increased in the base editor expressing SpCas9-NG (TAID-SpCas9-NG) with gRNA3 but not in the wild-type SpCas9 (TAID-SpCas9) with gRNA3 (Figs. 5c, d). The next-generation sequencing analysis showed a 1.74 ± 0.31% correction of the mutation in the liver after the vector administration (Fig. 5e).

## Discussion

The CRISPR-Cas9 genome-editing technologies are expected to be useful for the treatment of genetic diseases. However, attention has recently been directed to non-cutting genome editing, such as base editing[13,14], due to the genomic toxicity derived from DSBs[8–11]. A large percentage of genetic diseases are associated with point mutations[34,35], and around 60% of the disease-causing point mutations could be repaired by C-to-T or A-to-G base editing[36]. Point mutations are the most common type of mutation in hemophilia, and hemophilia B has a particularly high percentage of point mutations, even in severe cases[37], making it a good target disease for base editing. We searched the mutations of hemophilia B in the database of the European Association for Haemophilia and Allied Disorders (EAHAD). We extracted 4657

mutations (excluding 56 polymorphisms) and found that 67.2% of them may be treated by base editing (point mutation: A>G, C>T, G>A, and C>T) (Supplementary Table 2). However, the application of base editing to treat disease mutation remains limited as Cas9 requires PAM sequences for target DNA recognition, and base editors can efficiently edit limited few bases in target sites[36]. The present study revealed that a point mutation in human hemophilia B could be repaired by base editing with SpCas9-NG, an engineered Cas9 with broad PAM flexibility.

The efficiency of base editing with SpCas9-NG to repair the mutation was more than twofold greater than that of wild-type SpCas9. SpCas9-NG can bind to the target DNA site by recognizing only one G nucleotide and could therefore cover four times as many mutation sites. The PAM flexibility of SpCas9-NG in base editing has been previously reported in plants and rabbits[38,39]. Our data demonstrated that SpCas9-NG is effective in repairing human disease mutations for treatment. xCas9, another engineered Cas9 with broad PAM flexibility, also showed effective repair of human disease mutations by base editing[40]. However, the efficiency of base editing by xCas9 was much lower than that of SpCas9-NG[21]. Another variant of Cas9, SpRY, which requires only a single base as the PAM sequence (NRN and NYN), was recently developed[41]. These SpCas9 variants with high PAM flexibility exhibit lower editing efficiency compared with wild-type SpCas9 at target sites with NGG PAM sequences[7]. Indeed, we observed that SpCas9-NG induced DSBs at the gRNA-targeting NGG PAM site much less efficiently as compared with wild-type SpCas9. We replaced the Arg1335 of SpCas9, which recognizes the third "G" base, with Ala in SpCas9-NG to remove the base-specific interaction with this base[21]. The introduction of the mutation may expand the flexibility, although it may decrease the recognition against "NGG". Thus, it is crucial to select wild-type or engineered Cas9 with broad PAM flexibility according to the individual PAM sequence.

TAID showed a higher base-editing efficiency compared with BE4 for the causative mutation (c.947T>C) in a hemophilia B patient. The effective target range of base editing differs between TAID and BE4, and TAID and BE4 effectively induced C to T conversion 2–4 and 4–8 bp downstream of the 5′ end of the gRNA sequence, respectively[13,14]. When SpCas9-NG binds to the gRNA3-targeting DNA site, the causative mutation exists at the efficient base-editing range of TAID. Although wild-type SpCas9 efficiently induced DSBs at the gRNA2-targeting site and the causative mutation exists in the base-editing range of BE4 in the case of gRNA2, we failed to observe efficient editing by BE4. These data are consistent with previous reports showing that the base-editing efficiency of TAID near the 5′ end of gRNA sequence is higher than that of BE4, and the editing efficiency of BE4 differs between iPSCs and HEK293T cells[23,42]. As we could not precisely expect a successful outcome from the application of base-editing technology, comparing several combinations of base editors and Cas proteins was necessary to find an effective way to repair particular mutations. Recently, cytidine base editors with improved editing area and efficiency and new base editors, which can change C-to-A or C-to-G in mammalian cells, have been reported[43–46]. It may become possible to develop personalized genome-editing therapies by combining an engineered Cas9 with broad PAM flexibility and a variety of base-editing tools.

The present study has several limitations. First, the base-editing efficiency for the mutation of iPSC in this study was lower than that in previous reports[21]. One explanation for the low efficiency could be that the mutation site is difficult to repair by base editing. We predicted the efficiency of the base-editing approach using a Web software based on BE4max (http://deepcrispr.info/DeepBaseEditor/)[42] and found that the repair efficiency of the mutation was 12%. Since the base-editing efficiency of BE4 was

lower than that of BE4max[43], the mutation repair was expected to be less than 7%. It has also been reported that the base-editing efficiency in iPSCs did not match the predicted efficiency in HEK293 cells[42]. The application of improved base-editing tools would efficiently repair the mutation[43,44]. Second, there may have been a bystander effect of the base editor, in which the base-editing tool creates unwanted C to T alterations when more than one C is present in the editing window. We observed several unexpected C to T conversions near the target site (Supplementary Fig. 5a, b). Efforts to minimize bystander and off-target activities are required for future clinical application. Recent studies reported other genome-editing technologies that do not induce DSBs, such as prime editing, efficient homology-directed gene repair, and genome insertion by piggyback[47–50]. It will be interesting to compare the treatment effects of base editing with these other approaches. Additionally, we solely assessed the mutation derived from only one patient. Therefore, further investigations are required to validate the efficacy of base editing using SpCas9-NG. Finally, we were able to detect the expression of *F9* mRNA and FIX protein in the gene-corrected iPSC-derived cells but did not detect an increase in plasma FIX antigens. We suppose one possible reason why plasma FIX antigen could not be detected was an insufficient number of transplanted cells derived from the iPSCs in mice and incomplete differentiation. A highly efficient method for differentiating iPSCs into mature hepatocytes for therapeutic application remains to be established[51]. The advancement of novel technologies, including three-dimensional culture and organoid formation[52,53], may also lead to iPSC-based cell therapy for inherited liver diseases. We found that utilizing the AAV vector-mediated base-editing approach using SpCas9-NG increased plasma FIX in vivo but failed to completely normalize plasma FIX. Thus, we need to further optimize the efficiencies of the in vivo base-editing treatment. Alternatively, efficient genome editing in the liver hepatocyte could be achieved by lipid nanoparticle (LNP)-based intravenous delivery of Cas9 mRNA in a human clinical trial[54]. The delivery of CRISPR base editors using LNP could reportedly modify disease-related genes in cynomolgus monkeys in vivo[55]. The LNP-based approach combined with SpCas9-NG would facilitate in vivo base-editing therapy targeting various point mutations of inherited disorders.

In conclusion, we showed that PAM-flexible SpCas9-NG-mediated base editing effectively repairs point mutations in hemophilia B patient-derived iPSCs, as well as HEK29 cells and knock-in mice carrying the patient's mutation. The broad PAM flexibility of SpCas9-NG allows base editing at target sites that cannot be edited by wild-type SpCas9. Hence, base-editing technology based on SpCas9-NG can be used to treat point mutations in hereditary diseases. Further development of the genome-editing tool with high editing efficiency and specificity, as well as efficient delivery methods, may lead to personalized treatment by base editing for human genetic diseases.

## Data availability
Source data underlying the figures are provided in the Supplementary Data 2 file. The data obtained from the current study are available from the corresponding author upon reasonable request.

## Materials availability
All plasmids and cells are available from the corresponding author upon reasonable request.

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

## Acknowledgements

The present study was supported by the grant JP20fk0410017 (T.O.), JP21fk0410037 (T.O.), JP21ae0201007 (T.O.), JP20bm0804018 (H.U.), and JP20am0401005 (O.N.) from the Japan Agency for Medical Research and Development (AMED); The Access to Insights Basic Research Grant (T.O.); SENSHIN Medical Research Foundation (T.H.); Jichi Medical University Young Investigator Award (T.H.); and Takeda Science Foundation (T.H.). The authors would like to thank Yuiko Ogiwara, Yaeko Suto, Mika Kishimoto, Tamaki Aoki, Hiromi Ozaki, Mai Hayashi, Tomoko Noguchi, Nagako Sekiya, and Hiroko Hayakawa (Jichi Medical University) for their technical assistance and Daisuke Motooka (Osaka University) for next-generation sequencing and GUIDE-Seq analysis. Part of this manuscript was presented at the Congress of the International Society on Thrombosis and Haemostasis, July, 2020. BD LSRFortessa (BD Biosciences), a flow cytometer used in this study, was subsidized by JKA through its promotion funds from KEIRIN RACE.

## Author contributions

T.H. and T.O. designed the study, performed experiments, analyzed data, and wrote the manuscript; Y.K., N.B., H.I., and T.A. performed experiments and analyzed data; M.H. and N.K. helped in data analysis and revised manuscript; H.N., H.U., and O.N. provided critical reagent and analyzed data, and revised manuscript; Y.H. analyzed data and revised manuscript.

## Competing interests

T.O. received grant support from Novo Nordisk (*The Access to Insights* Basic Research Grant). No competing interest exists for the remaining authors.

**Additional information**

