## [Peer Review File · Communications Medicine]

Reviewers' comments:

Reviewer #1 (Remarks to the Author):

The work by Hiramoto et al described a series of experiments focused on the use of base editing coupled with a broad PAM sequence, thus the goal is to increase the desired correction without/minimal DSB.

The disease choice is hemophilia B due to mutations in the F9 gene, an ideal model for gene editing since the plethora of underlying mutations described worldwide (>1000 mutations) and relative straightforward assessment of the correction/improvement of the phenotype (FIX is a circulating protein and easily detected by multiple assays. The use of iPSC to generate hepatocyte is appropriate and the in vivo model tested is also adequate.

Although the experiments performed were correct, the novelty of this study is limited by the relative poor efficacy as noted by inability to detect the therapeutic protein in plasma samples. This makes this strategy unlikely to be of relevant to the field in terms of providing any feasibility, at this point.

Comments:

The main question is whether the PAM-modified was really effective. Overall, these data are in the right direction but far to support the claims in the section Discussion.

Figure 1 Panel E. The authors should comments on the relative efficiency observed between gRNA 2 by SpCas9 and SpCas9=NG. What was the determinant of the efficacy detected? In other systems: are these efficiencies of any relevance?

Figure 2 Panel B. The rationale of the better performance of TAID and BE4 is not fully developed. The authors should elaborate the differences between these models and the expected outcomes.

The Discussion should be focused on the data explored in this manuscript. The parts related to the translational potential such as the vector and etc for in vivo is not relevant at this point.

The Discussion is also too long and unfocused.

Reviewer #2 (Remarks to the Author):

The aim of this manuscript by Hiramoto et al. is to reverse a hemophilia B mutation in patient-derived iPSCs by base editing. The authors use spCas9-NG and deaminases TAID to correct the c.947T>C mutation and recover the F9 transcription. This manuscript performed comprehensive study and presented well. Nevertheless, there still a couple points that require addressed in the manuscript before its publication in Communications Medicine.

1. Why conjugated D10A Cas9 with TAID and APOBEC1 at C- and N- terminal, respectively?
2. To assess the safety of this approach, the off-target should be carefully analyzed.
3. WB or IHC analysis of the expression of FIX better in protein level is expected.
4. To highlight the significance of this approach, the authors should list the hemophilia B mutations which can be corrected by base editing.
5. There is no scale bar label in Figure 3b.

Reviewer #3 (Remarks to the Author):

The manuscript of Hiramoto and colleagues describes an improved SpCas9 for the efficient base-editing. As mentioned by the authors as well, base-editing is a promising approach for gene therapy, directly interfering with point mutations, without the need for DSB, which is clearly in advantage in the field. The authors have used their system in the context of Hemophilia B and FIX deficiency.

The manuscript is very nicely written and the respective figures are of very high quality, which makes easy for the reader to follow.

The overall statement of the manuscript is solid and justified. I only have some concerns about the in vivo data and respective expression levels of FIX in base-edited cells. Please find below my comments which should be addressed prior publication.

Major

1. The authors transplanted base-edited hepatic like cells in vivo to evaluate FIX expression. Thus, fig 3d and 3e are of great value for the manuscript. The authors should show the values of FIX plasma levels in transplanted animals.
2. Along this line, the authors could only detect FIX in the transplanted tissue (adjacent to the kidney). While I appreciate expression levels, I would suggest to perform tissue section and immunohistochemistry to allocate FIX expression to the respective transplanted, hepatic-like cell type.
3. In fig 3e, what does ex vivo mean? Would this relate to fig 3c, and if yes, how do the authors explain differences.
4. Overall, the authors should maybe perform one downstream assay of FIX demonstrating that the coagulation pathway is restored. Would this be possible?

Minor

1. Please harmonize the efficacy data of figure 1d and e showing the overall efficacy data of gRNA1 also in Fig 1E
2. Section of Figure 1: why do the authors cite ref 21 here? Has the very same construct used before in this study? Or is this rather a general statement about the flexibility of SpCas9-NG?
3. Have the authors performed off-target efficacy studies?
4. Just of curiosity: The authors mention the slight discrepancy in the efficacy using the different gRNAs in combination with TAID or BE4 (fig 2b to fig 1e). Do the authors used WT SpCas9 in combination with gRNA3 (plus TAID or BE4)? Although this combination performed poor in the T7 assay, it may be better if combined with TAID or BE4 on base editing?
5. I appreciate the expression levels in fig 3c, but how do these levels correlate to primary

cells? A positive control should be added to have a better impression of the maturity of cells.

6. Along this line, I would also incorporate a negative control, which should be the parental iPSC line with non-effective base-editing.

7. The discussion is very long in comparison to the overall manuscript. I would suggest to shorten the discussion to points really associated to the manuscript. E.g. parts for AAV etc could be deleted.

Point-by-point responses to the Reviewers' comments

We are grateful to the reviewers for their comments and helpful suggestions, which helped us to considerably improve our manuscript. Our point-by-point responses to the comments are presented below. We addressed all comments and suggestions in the revised version of our manuscript.

Responses to the comments of Reviewer 1

The work by Hiramoto et al described a series of experiments focused on the use of base editing coupled with a broad PAM sequence, thus the goal is to increase the desired correction without/minimal DSB.

The disease choice is hemophilia B due to mutations in the F9 gene, an ideal model for gene editing since the plethora of underlying mutations described worldwide (>1000 mutations) and relative straightforward assessment of the correction/improvement of the phenotype (FIX is a circulating protein and easily detected by multiple assays. The use of iPSC to generate hepatocyte is appropriate and the in vivo model tested is also adequate.

Although the experiments performed were correct, the novelty of this study is limited by the relative poor efficacy as noted by inability to detect the therapeutic protein in plasma samples. This makes this strategy unlikely to be of relevant to the field in terms of providing any feasibility, at this point.

Response: Thank you very much for your valuable comments and feedback regarding our manuscript. Each of your insight has served to strengthen our manuscript and we have made changes to reflect them. To strengthen our method to restore the disease phenotype, we employed HEK293 cells expressing the mutation. We confirmed the restoration of FIX activity and antigen in the supernatant of the cells after base editing (Figure 4).

Comment 1: The main question is whether the PAM-modified was really effective. Overall, these data are in the right direction but far to support the claims in the section Discussion.

Response: We have previously shown that C to T base editing using SpCas9-NG conjugated with TAID, but not wild type SpCas9, was possible in several target sites other than NGG (Nishimasu H, *et al.*, Science 2018, Figure 4). Although we assessed only one mutation from the patient in this manuscript, we believe the base-editing approach using SpCas9-NG has a great potential to modify the various disease point mutation. We added the following explanation in a limitation; **We only assessed the mutation derived from only one patient. Further investigations are required to validate the efficacy of base-editing using SpCas9-NG (Page 9, lines 32-33).**

Comment 2: Figure 1 Panel E. The authors should comment on the relative efficiency observed between gRNA 2 by SpCas9 and SpCas9=NG. What was the determinant of the efficacy detected? In other systems, are these efficiencies of any relevance?

Response: The PAM sequence for gRNA2 is “NGG,” which is efficiently recognized with wild-type SpCas9. We showed that the cleavage by SpCas9-NG at the target site with the “NGG” PAM was weaker than that by SpCas9. This result is consistent with the findings reported in our previous study (Nishimasu et al., 2018, published in Science), indicating that the cleavage efficiency by SpCas9-NG at the NGG target was lower than that by wild-type SpCas9. We replaced the Arg1335 of SpCas9, which recognizes the third “G” base, with Ala to remove the base-specific interaction with this base. The introduction of the mutation may expand the flexibility, although it may decrease the recognition against “NGG.” These observations were incorporated into the revised manuscript (Page 8, line 27– Page 9, line 1).

Comment 3: The rationale of the better performance of TAID and BE4 is not fully developed. The authors should elaborate the differences between these models and the expected outcomes.

Response: Both the TAID and BE4 base editing tools are capable of C-to-T base editing, but their editing windows are different (i.e., TAID-SpCas9: two to four bases from 5’ end; BE4-SpCas9: four to eight bases from 5’ end). While the editing windows of TAID (but not BE) include the mutation at gRNA3, it is expected to edit the mutation by BE at gRNA2. However, we can observe the efficient restoration of the mutation only by combining gRNA3 and TAID-SpCas9-NG but not by combining gRNA2 and BE-SpCas9. These findings are consistent with those reported in previous studies showing that the base-editing efficiency of TAID near the 5’ end of the gRNA sequence is higher than that of BE4 and that the editing efficiency of BE4 differs between iPSCs and HEK293T cells. As we could not precisely expect a successful outcome from the application of base editing technology, comparing several combinations of a base editor with Cas proteins was necessary to find an effective way to repair particular mutations. These observations were incorporated into the revised manuscript (Page 9, lines 3 –14).

Comment 4: The Discussion should be focused on the data explored in this manuscript. The parts related to the translational potential such as the vector and etc for in vivo is not relevant at this point.

The discussion is also too long and unfocused.

Response: As advised, we shortened the Discussion section.

Responses to the comments of Reviewer 2

The aim of this manuscript by Hiramoto et al. is to reverse a hemophilia B mutation in patient-derived iPSCs by base editing. The authors use spCas9-NG and deaminases TAID to correct the c.947T>C mutation and recover the F9 transcription. This manuscript performed comprehensive study and presented well. Nevertheless, there still a couple points that require addressed in the manuscript before its publication in Communications Medicine.

Response: Thank you very much for your valuable comments and feedback regarding our manuscript. Each of your insight has served to strengthen our manuscript and we have made changes to reflect them.

Comment 1: Why was conjugated D10A Cas9 with TAID and APOBEC1 at C- and N- terminal, respectively?

Response: We conjugated TAID or BE4 at the same position of SpCas9, based on methods employed in previous studies (Nishida et al., 2016, published in Science; Komor et al., 2017, published in Sci. Adv.). We cited these papers in the revised manuscript (Page 11, lines 23–25).

Comment 2: To assess the safety of this approach, the off-target should be carefully analyzed.

Response: We could not directly assess the genome-wide off-target events by base editing. Therefore, as an alternative, we evaluated the genome-wide cleavage events of SpCas9-NG with gRNA3 compared with wild-type SpCas9 by GUIDE-Seq. The peak score was significantly higher for SpCas9-NG at one guide RNA mismatch, which corresponds to the point mutation. We could not detect any difference between the other scores and influenced genes of SpCas9 and those of SpCas9-NG (Supplemental Figure 1, Supplemental Table 1; Page 5, lines 20–25).

Comment 3: WB or IHC analysis of the expression of FIX better in protein level is expected.

Response: Because we could not detect the human FIX antigen in the gene-corrected iPSCs-engrafted mouse plasma, we established the HEK293 cells that expressed the *F9* cDNA with the patient's mutation. HEK293 cells harboring the mutated *F9* cDNA (I316T mutation) expressed the *F9* mRNA and FIX protein in the cytoplasm but did not express the FIX antigen and FIX activity in the supernatant, suggesting that the I316T mutation abolishes the extracellular export of FIX. Next, we transfected HEK293 cells expressing the mutated *F9* cDNA with TAID-SpCas9-NG and gRNA3. We observed that the FIX activity in the supernatant was restored in eight out of 127 clones, confirming the proper base-editing of *F9* cDNA. These results were incorporated into the revised manuscript (Figure 4 and Supplemental Figure 3; Page 7, lines 5-19).

Comment 4: To highlight the significance of this approach, the authors should list the hemophilia B mutations which can be corrected by base editing.

Response: We searched the mutations of hemophilia B in the database of the European Association for Haemophilia and Allied Disorders (EAHAD). We extracted 4,657 mutations (excluding 56 polymorphisms) and found that 67.2% of them may be treated by base editing (point mutation: A > G, C > T, G > A, and C > T). The results were incorporated into the revised manuscript (Supplemental Table 2; Page 8, lines 8–11).

Comment 5: There is no scale bar label in Figure 3b.

Response: We incorporated the scale bar in Figure 3b.

Responses to the comments of Reviewer 3

The manuscript of Hiramoto and colleagues describes an improved SpCas9 for the efficient base-editing. As mentioned by the authors as well, base-editing is a promising approach for gene therapy, directly interfering with point mutations, without the need for DSB, which is clearly in advantage in the field. The authors have used their system in the context of Hemophilia B and FIX deficiency.

The manuscript is very nicely written and the respective figures are of very high quality, which makes easy for the reader to follow.

The overall statement of the manuscript is solid and justified. I only have some concerns about the in vivo data and respective expression levels of FIX in base-edited cells. Please find below my comments which should be addressed prior publication.

Response: Thank you very much for your valuable comments and feedback regarding our manuscript. Each of your insight has served to strengthen our manuscript and we have made changes to reflect them.

Major:

Comment 1: The authors transplanted base-edited hepatic like cells in vivo to evaluate FIX expression. Thus, fig 3d and 3e are of great value for the manuscript. The authors should show the values of FIX plasma levels in transplanted animals.

Response: The FIX antigen in the gene-corrected iPSC-engrafted mouse plasma was hardly detected. Therefore, as an alternative, we established HEK293 cells that expressed the *F9* cDNA with the patient's mutation (I316T mutation) to show the restoration of the disease phenotype by base editing. The HEK293 cells expressing *F9* with the I316T mutation expressed the *F9* mRNA but did not produce the FIX antigen and corresponding activity in the culture media. We repaired the mutation by transducing the cells with TAID-SpCas9-NG and gRNA3. After the transduction, we confirmed the increase in FIX activity in the medium in eight out of 137 clones. We confirmed that the mutation was repaired in the clones with elevated FIX activity (Figure 4g). These results were incorporated into the revised manuscript (Figure 4; Page 7, lines 5-19).

Comment 2: Along this line, the authors could only detect FIX in the transplanted tissue (adjacent to the kidney). While I appreciate expression levels, I would suggest performing tissue section and immunohistochemistry to allocate FIX expression to the respective transplanted,

Response: As mentioned above in the response to comment 1, we established HEK293 cells that expressed the *F9* cDNA with the patient's mutation (I316T mutation). We found that the mutation abolished FIX production from the cells but did not suppress mRNA expression or protein production (Figure 4 and Supplemental Figure 3; Page 7, lines 5-19).

Comment 3: In fig 3e, what does ex vivo mean? Would this relate to fig 3c, and if yes, how do the authors explain differences.

Response: The expression “ex vivo” in Figure 3e indicated the samples derived from cells that were differentiated in vitro. Hence, “ex vivo” was changed to “in vitro” throughout the manuscript. Further, we explained the sample information in the legend of Figure 3 (Page 23, lines 17–18).

Comment 4: Overall, the authors should maybe perform one downstream assay of FIX demonstrating that the coagulation pathway is restored. Would this be possible?

Response: As mentioned above in the response to comment 1, we confirmed the restoration of FIX production by base editing in the HEK293 cells expressing the patient’s mutation (Figure 4; Page 7, lines 5-19).

Minor:

Comment 5: Please harmonize the efficacy data of figure 1d and e showing the overall efficacy data of gRNA1 also in Fig 1E

Response: We incorporated the results of SpCas9 and SpCas9-NG with gRNA1 in Figure 1e.

Comment 6: Section of Figure 1: why do the authors cite ref 21 here? Has the very same construct used before in this study? Or is this rather a general statement about the flexibility of SpCas9-NG?

Response: We deleted the citation in the revised manuscript.

Comment 7: Have the authors performed off-target efficacy studies?

Response: We could not directly assess the genome-wide off-target events by base editing. As an alternative, we evaluated the genome-wide cleavage events of SpCas9-NG with gRNA3 and compared them with wild-type SpCas9 by GUIDE-Seq. The peak score was significantly higher for SpCas9-NG at one guide RNA mismatch, which corresponds to the point mutation. We could not detect any difference between the other scores and influenced genes of SpCas9, and those of SpCas9-NG (Supplemental Figure 1, Supplemental Table 1; Page 5, lines 20–25).

Comment 8: Just of curiosity: The authors mention the slight discrepancy in the efficacy using the different gRNAs in combination with TAID or BE4 (fig 2b to fig 1e). Do the authors used WT SpCas9 in combination with gRNA3 (plus TAID orBE4)? Although this combination performed poor in the T7 assay, it may be better if combined with TAID or BE4 on base editing?

Response: We incorporated the data of gRNA3 with TAID- and BE4-conjugated SpCas9 in Figure 2. We could not detect base editing by gRNA3 with TAID- and BE4-conjugated SpCas9 (Page 6, lines 3–4).

Comment 9: I appreciate the expression levels in fig 3c, but how do these levels correlate to primary cells? A positive control should be added to have a better impression of the maturity of cells.

Response: We measured the expression levels in human liver RNA as the positive control. The expressions of *AFP*, *ALP*, and *CYP3A* in the cells that were differentiated in vitro were comparable to, or greater than, those in human liver; however, the expressions of *HNF4A* and *F9* were significantly lower than those in human liver RNA, suggesting insufficient differentiation of iPSCs into mature hepatocytes. The expression of *F9* did not reach the level in normal liver cells even after the in vivo differentiation. This explanation was incorporated into the revised manuscript (Supplemental Figure 2; Page 6, line 21- Page 7, line 3).

Comment 10: Along this line, I would also incorporate a negative control, which should be the parental iPSC line with non-effective base-editing.

Response: We presented the differentiation of parental iPSC data in Supplementary Figure 2. To demonstrate the correction of the disease phenotype by base editing technology, we confirmed the restoration of FIX production in HEK293 cells harboring the patient's *F9* cDNA.

Comment 11: The discussion is very long in comparison to the overall manuscript. I would suggest shortening the discussion to points really associated to the manuscript. E.g. parts for AAV etc could be deleted.

Response : As advised, we shortened the discussion section.

Reviewers' comments:

Reviewer #1 (Remarks to the Author):

The authors should explain in a clear and short statement the current stage of the proposed strategy to avoid the readers to think that these data are of translation potential. The data presented are likely to show a potential, yet to be optimized, strategy.

Reviewer #2 (Remarks to the Author):

In this revised manuscript, the authors addressed some of points raised by the reviewers. The manuscript has been improved. Nevertheless, I would like to see the following points to be addressed in the manuscript before its publication.

Comments/Suggestions,

1. The expression levels of F9 is about 1/10 of those observed in normal human liver (in vivo differentiated cells, $6.6 \pm 1.1 \times 10^4$ vs. normal human liver, $86.6 \pm 9.9 \times 10^4$), but the mice is about 1/100 of human size, please explain why human FIX antigen cannot be detected in the mice plasma.
2. Subrenal capsule grafting technology is more commonly used in cancer research. Please explain the reason for choosing this method.
3. To highlight the novelty of this study, better to show the protein level of F9 in animal model. Otherwise, the guiding significance of this study to clinical research will be greatly reduced.

Reviewer #3 (Remarks to the Author):

The authors have addressed the comments.

Point-by-point responses to the Reviewers' comments

We ardently thank the reviewers for their instructive comments and suggestions. We have addressed all the comments and suggestions in the revised version of our manuscript. Our point-by-point responses to the comments are as follows:

Responses to the comments of Reviewer 1

The authors should explain in a clear and short statement the current stage of the proposed strategy to avoid the readers to think that these are data are of translation potential. The data presented are likely to show a potential, yet to be optimized, strategy.

Response: We thank you for your insightful comments and feedback regarding our manuscript. As you highlighted, our data demonstrated the potential of SpCas9-NG for the repair of a point mutation by base editing; however, it only validated the repair of the mutation in the cell-based assay. For clinical application, it is necessary to evaluate how to transplant the cells derived from the gene-corrected induced pluripotent stem cells (iPSCs) or directly deliver genome editing tools to the liver. We have incorporated the limitation and discussed several future perspectives in the revised manuscript (Page 10, lines 4–17).

Responses to the comments of Reviewer 2

In this revised manuscript, the authors addressed some of points raised by the reviewers. The manuscript has been improved. Nevertheless, I would like to see the following points to be addressed in the manuscript before its publication.

Response: Thank you for your helpful comments and feedback regarding our manuscript. We agree with the reviewer's comment on the limitations of our research. Our data indirectly attributable to the treatment strategy of hemophilia and demonstrated the potential of SpCas9-NG for point mutation repair by base editing. For future clinical applications, it is necessary to evaluate how to transplant the cells derived from the gene-corrected iPSCs or directly deliver genome editing tools to the liver. We have incorporated these limitations and discussed future perspectives in the revised manuscript.

Comments/Suggestions,

1. The expression levels of F9 is about 1/10 of those observed in normal human liver (in vivo differentiated cells, $6.6 \pm 1.1 \times 10^4$ vs. normal human liver, $86.6 \pm 9.9 \times 10^4$), but the mice is about 1/100 of human size, please explain why human FIX antigen cannot be detected in the mice plasma.

Response: A possible reason why plasma coagulation factor IX (FIX) antigen were not detected in the plasma could be insufficient number of transplanted cells derived from the iPSCs in mice, and the incomplete differentiation of iPSCs. Mouse liver weight approximately 1.5 g in adult mice (5% body weight) (Nagatani M, *et al.* Experimental Animals, Vol. 68;2019, pp471-482). The number of hepatocytes in the liver was reportedly 135×10^6 cells/g (total 203×10^6 cells) (Sohlenius-Sternbeck, AK. Toxicology in Vitro, Vol. 20;2006, pp1582-1586). The number of transplanted cells was <1% of mouse liver (2×10^6 cells). If all differentiated cells were completely engrafted and produced FIX similar to liver hepatocytes, plasma FIX might not increase above 1%. These explanations were incorporated into the revised manuscript (Page 10, lines 4–11).

2. Subrenal capsule grafting technology is more commonly used in cancer research. Please explain the reason for choosing this method.

Response: As the reviewer indicated, the subrenal capsule transplantation procedure can be commonly used to graft human cancer cells, while several studies have reported the long-term engraftment of the transplanted hepatocytes or the differentiated cells from iPSCs (Zhang RR, *et al.* Stem Cell Reports., Vol. 10;2018, pp780-793, Koike H, *et al.* Nature., Vol. 574;2019, pp112-116, Harrison SP, *et al.* bioRxiv, 2020: 2020.12.02.406835), we employed the previous procedure for differentiating iPSCs and transplanting the cells (Okamoto R, *et al.* Cell Transplantation, Vol. 27;2018, pp299-309). The explanation is described in the revised manuscript (Page 6, line 27–Page 7, line 1).

3. To highlight the novelty of this study, better to show the protein level of F9 in animal model. Otherwise, the guiding significance of this study to clinical research will be greatly reduced.

Response: As you highlighted, our data demonstrated the potential of SpCas9-NG for the repair of a point mutation by base editing but indirectly provided the treatment strategy for hemophilia. Transplantation of the cells derived from gene-corrected autologous iPSCs may be a promising future strategy. However, a highly efficient method for differentiating iPSCs into mature hepatocytes for therapeutic application remains unestablished (Reza HA, Okabe R, and Takebe T. Transplant International, Vol. 34;2021, pp2031-2045). Recently, there have been a few reports on creating a liver structure using 3D culture (Luce E, *et al.* Hepatology. Vol. 75;2022, pp866-880) and organoid formation (Velazquez JJ. *et al.* Cell Systems, Vol. 12;2020, pp41-65). The improvement of these technologies will lead to iPSC-based cell therapy for inherited liver disorders. Alternatively, efficient genome editing in the liver hepatocytes could be achieved by lipid nanoparticle (LNP)-based intravenous delivery of Cas9 mRNA in a human clinical trial (Gillmore JD, *et al.* N Engl J Med, Vol. 385;2021, pp493-502). Clustered Regulatory Interspaced Short Palindromic Repeats (CRISPR) base editors using LNP could reportedly modify disease-related genes in cynomolgus monkeys *in vivo* (Musunuru K, *et al.* Nature, Vol. 593;2021, pp429-434). The LNP-based approach combined with SpCas9-NG would facilitate *in vivo* base editing therapy targeting various point mutations

of inherited disorders. These explanations are incorporated into the revised manuscript (Page 10, lines 4–17).

Reviewers' comments:

Reviewer #2 (Remarks to the Author):

The manuscript has been improved by addressing some points raised by the reviewers. However, I insisted the expression levels of F9 in animal model is critical for this research.

Additional comments from Reviewer 3:

I agree with the authors, that transplantation of iPSC-derived hepatocytes and subsequent engraftment is, at the current stage hard to achieve (and maybe also out of the scope of the manuscript). Thus, the authors used the transplantation under the kidney capsule, aiming to evaluate F9 expression in vivo. Of note, F9 expression in vivo could only be detected by mRNA levels and failed to be detected in the serum. Although I share the explanation given by the authors, the detection of F9 in vivo would be the most important goal and would definitely increase the overall impact of the manuscript.

Given the main scope of the manuscript (base editing) and the attempts of the authors to detect F9 expression in vitro and in vivo, a compromise for functional F9 expression could be:

1. data which proves really the functionality of the expressed F9 (see also R#3 comment 4). This data would accompany the already present data of the manuscript from a “functionality” point of view
2. As alternative, and maybe also more in line with R#2, the authors could perform tissue sections of the liver and kidney capsule using immunofluorescent staining for F9 (as done on HEK cells Fig4D)(see also R#3 comment 2). Given the low levels of F9 in vivo, such IF staining would underline the mRNA expression levels of F9 in vivo.

I hope that the two options raised before could be a compromise for R#2 highlighting the expression of F9, which I agree is an important part of the manuscript.

Below some additional literature for the functional assessment of F9.

<https://pubmed.ncbi.nlm.nih.gov/34687060/>

PhD thesis

<https://open.bu.edu/handle/2144/31237>

Point-by-point responses to the Reviewers' comments

We ardently thank the reviewers for their instructive comments and suggestions. We have addressed all the comments and suggestions in the revised version of our manuscript. Our point-by-point responses to the comments are as follows:

Responses to the comments of Reviewer 2

The manuscript has been improved by addressing some points raised by the reviewers. However, I insisted the expression levels of F9 in animal model is critical for this research.

Response: To address the Reviewer's comment regarding FIX in the animal model, we employed two approaches:

- (1) We performed immunostaining of the engrafted tissue derived from the gene-corrected iPSCs and detected FIX, suggesting that the engrafted tissue indeed produced FIX protein (see Fig. 3e in the revised manuscript).
- (2) We employed AAV vector-mediated base-editing in vivo. First, we created knock-in (KI) mice that carried the patient's mutation (Fig. 5a). To express the base editors in the liver, we utilized an AAV system that uses protein trans-splicing. We administrated the two AAV vectors intraperitoneally into newborn pups of KI mice, to express one base editor. We confirmed the increase in plasma levels of FIX and the gene correction by the base editor expressing SpCas9-NG (TAID-SpCas9-NG) with gRNA3 (Figs. 5c-e). As expected, using wild-type SpCas9 (TAID-SpCas9) with gRNA3 produced no discernible effects. We incorporated these data and related limitations into the revised manuscript (Fig. 5; Page 7, line 25 – Page 8, line 12; Page 11, lines 4–13).

Responses to the Additional comments of Reviewer 3

I agree with the authors, that transplantation of iPSC-derived hepatocytes and subsequent engraftment is, at the current stage hard to achieve (and maybe also out of the scope of the manuscript). Thus, the authors used the transplantation under the kidney capsule, aiming to evaluate F9 expression in vivo. Of note, F9 expression in vivo could only be detected by mRNA levels and failed to be detected in the serum. Although I share the explanation given by the authors, the detection of F9 in vivo would be the most important goal and would definitely increase the overall impact of the manuscript.

Given the main scope of the manuscript (base editing) and the attempts of the authors to detect F9 expression in vitro and in vivo, a compromise for functional F9 expression could be:

1. data which proofs really the functionality of the expressed F9 (see also R#3 comment 4). This data would accompany the already present data of the manuscript from a “functionality” point of view

2. As alternative, and maybe also more in line with R#2, the authors could perform tissue sections of the liver and kidney capsule using immunofluorescent staining for F9 (as done on HEK cells Fig4D)(see also R#3 comment 2). Given the low levels of F9 in vivo, such IF staining would underline the mRNA expression levels of F9 in vivo.

I hope that the two options raised before could be a compromise for R#2 highlighting the expression of F9, which I agree is an important part of the manuscript.

Below some additional literature for the functional assessment of *F9*.

<https://pubmed.ncbi.nlm.nih.gov/34687060/>

PhD thesis

<https://open.bu.edu/handle/2144/31237>

Response: We thank you for your insightful comments and feedback. In response, we differentiated the gene-corrected iPSCs into the hepatoblasts and transplanted the cells into hemophilia B mouse pups using the same method as the suggested paper (Luce E. et al., *Hepatology*. 2022). However, we could not detect plasma FIX activities in the transplant-treated mice (data not shown). Although it is not clear why the data could not be reproduced, one explanation may be that a cassette composing the *F9* gene driven by hepatocyte-specific promoter into the *AAVSI* locus (rather than the physiological *F9* locus) was used in the referenced literature. As you suggested, we also performed immunostaining of the engrafted tissue and could indeed detect FIX expression (Fig. 3e in the revised manuscript). Since we agree that the increase in plasma FIX from the base-editing approach significantly enhances the impact of our paper, we employed the AAV vector-mediated base-editing approach, generating knock-in (KI) mice harboring the patient's mutation. To express the base editors in the liver, we applied an AAV system using protein trans-splicing. We administrated the two AAV vectors intraperitoneally to express one base editor into newborn KI mice pups. We confirmed the increase in FIX plasma levels and the gene correction by the base editor expressing SpCas9-NG (TAID-SpCas9-NG) with gRNA3 (Figs. 5c–e). As expected, using wild-type SpCas9 (TAID-SpCas9) with gRNA3 was unsuccessful. We incorporated these data and related limitations into the revised manuscript (Fig. 5; Page 7, line 25 – Page 8, line 12; Page 11, lines 4–13).

REVIEWERS' COMMENTS:

Reviewer #2 (Remarks to the Author):

The authors have adequately addressed all my concerns. The manuscript has been substantially improved, and is suitable for publication in Communications Medicine.

Reviewer #3 (Remarks to the Author):

I thank the authors for the revised manuscript and dataset provided.

Point-by-point responses to the Reviewers' comments

REVIEWERS' COMMENTS:

Reviewer #2 (Remarks to the Author):

The authors have adequately addressed all my concerns. The manuscript has been substantially improved, and is suitable for publication in Communications Medicine.

Reviewer #3 (Remarks to the Author):

I thank the authors for the revised manuscript and dataset provided.

Response: We really appreciate all reviewers for their insightful comments and feedback regarding our manuscript. The manuscript has been improved after the revision. Thank you.